# Cytosolic Hsp90 Isoform-Specific Functions and Clinical Significance

**DOI:** 10.3390/biom12091166

**Published:** 2022-08-23

**Authors:** Samarpan Maiti, Didier Picard

**Affiliations:** Département de Biologie Moléculaire et Cellulaire, Université de Genève, Sciences III, Quai Ernest-Ansermet 30, CH-1211 Geneve, Switzerland

**Keywords:** molecular chaperone, paralog, Hsp90 isoforms, Hsp90α, Hsp90β, Hsp90-isoform specific inhibitors, clinical relevance

## Abstract

The heat shock protein 90 (Hsp90) is a molecular chaperone and a key regulator of proteostasis under both physiological and stress conditions. In mammals, there are two cytosolic Hsp90 isoforms: Hsp90α and Hsp90β. These two isoforms are 85% identical and encoded by two different genes. Hsp90β is constitutively expressed and essential for early mouse development, while Hsp90α is stress-inducible and not necessary for survivability. These two isoforms are known to have largely overlapping functions and to interact with a large fraction of the proteome. To what extent there are isoform-specific functions at the protein level has only relatively recently begun to emerge. There are studies indicating that one isoform is more involved in the functionality of a specific tissue or cell type. Moreover, in many diseases, functionally altered cells appear to be more dependent on one particular isoform. This leaves space for designing therapeutic strategies in an isoform-specific way, which may overcome the unfavorable outcome of pan-Hsp90 inhibition encountered in previous clinical trials. For this to succeed, isoform-specific functions must be understood in more detail. In this review, we summarize the available information on isoform-specific functions of mammalian Hsp90 and connect it to possible clinical applications.

## 1. Introduction

Heat shock proteins (Hsps) are molecular chaperones which are known for their numerous roles in protein homeostasis (proteostasis), including protein folding and refolding, maturation, disassembly of aggregates, and degradation [1,2,3]. The term “heat shock” proteins was coined as a legacy of Ritossa’s pioneering discovery that heat shock produced chromosomal puffs in the salivary glands of *Drosophila* larvae [4,5]. Later, it was established that the heat-shock response (HSR) is a universal response to an extensive array of stresses [6,7]. HSPs are not only essential during stress, but they are equally crucial in normal conditions to maintain proteostasis [8,9]. The human genome organization (HUGO) gene nomenclature committee recognizes five human HSP families (http://www.genenames.org/data/genegroup/#!/group/582; accessed on 5 August 2022) based on their observed molecular weights: Hsp70, Hsp90, Hsp40, the small HSPs, and chaperonins [10,11]. All Hsp90s consist of three major domains: an N-terminal ATPase domain (NTD), which binds ATP, a middle domain (MD), to which perhaps most clients bind, and a C-terminal dimerization domain (CTD) [12]. In this review, we will focus on mammalian Hsp90 and its cytosolic isoforms.

## 2. Hsp90 Homologs and Paralogs

Homologous genes or proteins (homologs) are genes or proteins of different species with a common ancestor, whereas paralogs are genes with sequence homology that originate from the intragenomic duplication of an ancestral gene. For the Hsp90 family of proteins, there are homologs in all organisms except in *Archaea* and some bacterial species [13,14]. Hsp90 is conserved from bacteria to humans with a sequence homology of about 53% between *Escherichia coli* (*E. coli*) and humans, which is a strong indication that this protein has remained vital throughout evolution [15]. During evolution, gene duplications allowed the divergence into different paralogous Hsp90 genes, which encode protein isoforms [14,16,17,18]. These additional isoforms, including organelle-specific ones, with different functional properties evolved as organisms gained more complexity [13,14,16,17,18,19]. Bacteria generally have only one isoform, known as the high-temperature protein G (HtpG) [19], but some bacterial species, such as *Streptomyces coelicolor*, contain another paralog, which shares only 30% identity [14]. In some bacteria, HtpG is essential during heat stress [20]. In *E. c**oli*, although HtpG is not essential, it is relatively abundant under non-stress conditions and further induced during heat stress [21], and indeed, *E. coli htpG* mutants have a growth defect at higher temperatures [22]. During evolution of eukaryotes, Hsp90 gained more importance and became an essential protein for viability [14,18]. Its importance is further emphasized by its abundance. Hsp90 comprises 1–2% of the total cellular protein in unstressed cells and up to 4–6% in the presence of stress [23,24]. The Hsp90 chaperone machinery is a key regulator of proteostasis, both in normal and stress conditions in eukaryotic cells [15,24,25]. In the unicellular eukaryote *Saccharomyces cerevisiae* (*S. cerevisiae*), genome-wide studies suggested that up to 10% of all proteins are directly or indirectly dependent on Hsp90 for function [26,27]. *S.*
*cerevisiae* possesses two cytosolic Hsp90 isoforms encoded by separate genes, which arose from a genome duplication: the cognate Hsc82 and the stress-inducible Hsp82 [28]. Under non-stress conditions, Hsc82 is expressed at tenfold higher levels than Hsp82 [28]. During heat shock, a strong induction of Hsp82 and a merely moderate induction of Hsc82 almost equalize the levels of the two isoforms. Hsp82 and Hsc82 are 709 and 705 amino acids long, respectively, with 96% identity and only 27 amino acid differences [13,29]. In multicellular organisms, there are four different types of Hsp90 paralogs based on their organelle-specific localization. These are the cytosolic Hsp90s [30,31,32,33], Trap1 in mitochondria [34,35], Grp94 in the endoplasmic reticulum [36], and chloroplast Hsp90C in plants [37]. Although these paralogs share many highly conserved domains with over 50% sequence identity, they differ in their functions [14,16,17,18]. In this review, we will focus on the cytosolic Hsp90 isoforms of mammals.

## 3. Cytosolic Hsp90

In mammals, there are two cytosolic Hsp90 isoforms: Hsp90α and Hsp90β [31]. Human Hsp90α and Hsp90β are encoded by the *HSP90AA1* and *HSP90AB1* genes, respectively [31,32,33]. Hsp90α was the first Hsp90 to be purified from heat-stressed Hela cells [32]. Later, Hsp90β was cloned based on homology to Hsp90α [33]. Millions of years ago, Hsp90α and Hsp90β originated by gene duplication [38]. These two cytosolic Hsp90 isoforms are highly homologous, with about 84% sequence identity (for 732 and 724 amino acids, respectively) [33]. There are five highly conserved signature sequences. Three are in the N-terminal domain and two are in the middle domain, comprising amino acids 38–59, 106–114, and 130–145, and 360–370 and 387–401, respectively [39]. What makes these two isoforms different in structure is that Hsp90α contains the 9-amino acid extension TQTQDQPME within the very N-terminal residues 4 to 12, which is replaced in Hsp90β by the 4-amino acid segment VHHG [31]. Hsp90β also has the unique signature sequence LKID (residues 71–74), which is not present in any other HSP [14]. Both isoforms function as homodimers [40,41]. Interestingly, the relatively poor dimer formation of human Hsp90β can be mapped to two amino acid differences compared to Hsp90α [42]. While there is evidence for vertebrates that some isoform heterodimers exist as well [43,44], mass spectrometric analysis revealed no α-β heterodimers [45] and sepharose-immobilized Hsp90β pulls out only β [46]. Whereas in *S. cerevisiae*, Hsp90 isoforms readily form heterodimers both in vitro and in vivo [29], this is disfavored in humans [47]. Hence, it remains to be investigated to what extent cytosolic isoform heterodimers exist in multicellular organisms, what regulates the equilibrium between homodimers and heterodimers, and whether they have distinct functions [12,39] (Figure 1).

### 3.1. Tissue-Specific Expression

The expression levels of the two isoforms varies in a tissue-specific way. In mice, Hsp90β is highly abundant in heart, liver, spleen, lung, intestine, muscle, brain, testis, and kidney. In comparison, the levels of Hsp90α are lower in those tissues. However, in testis, retina, and brain, Hsp90α levels are comparatively higher than Hsp90β, whereas in heart and muscle, Hsp90α is almost absent [48,49]. Indeed, we demonstrated an interesting isoform switch in mouse myoblasts: as they differentiate into myotubes, Hsp90α disappears and only Hsp90β remains [50] (see also chapter 3.2). According to the human protein atlas (https://www.proteinatlas.org; accessed on 5 August 2022), human brain has the highest expression of Hsp90α mRNA, and yet, this is not reflected at the protein level, as it is only moderate. For Hsp90β, there appears to be no such disparities (https://www.proteinatlas.org/search/HSP90AB1; accessed on 5 August 2022). As in mice, human Hsp90β protein is moderately to highly expressed in all the major tissues. Hsp90α protein is more highly expressed in the respiratory system, and in female and male reproductive organs. These differences in tissue distribution patterns suggest that different tissues have distinct isoform-specific functional requirements. It is intriguing to speculate that different tissues might have different intrinsic levels of biophysical stress, due to differences in temperature, osmotic pressure, and oxygen availability, which both affect the differential expression of the two isoforms and impose distinct functional requirements.

### 3.2. Isoform-Specific Co-Chaperones

During evolution from prokaryotes to eukaryotes, overall proteome complexity dramatically increased without any accompanying gain of core molecular chaperones. To assist the core molecular chaperones and to diversify their functions, a large panel of co-chaperones appeared [16]. Only relatively few co-chaperones are required for core Hsp90 functions (Table 1). These co-chaperones help in the transfer of clients from Hsp70, the N-terminal closure of the client-bound Hsp90 dimer, stimulating/inhibiting the ATPase activity, and the maturation of clients [51,52]. Other co-chaperones have more specialized functions suggesting a correlation between client diversity and the range of available co-chaperones. For now, there is still only limited evidence for isoform specificity of co-chaperones. In a recent review, Dean and Johnson discussed the relative expression of co-chaperones across a wide range of tissues [53]. The results of this survey suggest that some Hsp90 co-chaperones are uniquely required to assist client proteins in certain tissues. For example, the levels of the mRNAs encoding FKBP51, S100A1, ITGB1BP2, Unc45B, Aarsd1, and Harc are elevated in skeletal muscle, and most of them are also elevated in the heart. This correlative observation suggests that these co-chaperones may have muscle-specific functions [53], possibly with a corresponding Hsp90 isoform preference. ITGB1BP2 binds integrin and regulates the interaction between the cytoskeleton and the extracellular matrix, which have cardioprotective effects [54]. Unc45B assists the folding of myosin [55] and S100A1 is a regulator of muscle contractility [56]. FKBP51 functions in myoblast differentiation and in regulating muscle mass [57,58]. We showed that as myoblasts differentiate into myotubes, the co-chaperone p23 is replaced by the muscle-specific co-chaperone Aarsd1, which shares the Hsp90-interacting CS domain with p23, but not all of its activities. We found that the long isoform Aarsd1L interacts exclusively with Hsp90β, the only remaining and functionally important Hsp90 isoform in murine myotubes [50]. An inverse situation may pertain to spermatogenesis where Hsp90α levels are high. This isoform is essential [59], and a subset of co-chaperones have been linked to spermatogenesis. This includes PIH1D1, PIH1D2, PIH1D3, RPAP3, SPAG1, DYX1C1, LRRC6, and NUDCD1 [60,61]. Moreover, loss of FKBP36 results in chromosome mispairing during meiosis and mutations are suspected to cause azoospermia [62,63]. These examples illustrate that expression of Hsp90 isoforms, co-chaperones, and clients, and their respective interactions may have evolved to provide a unique match in certain tissues and cell types.

### 3.3. Isoform-Specific Post-Translational Modifications

Post-translational modifications can have a large impact on the function and regulation of the two isoforms [64]. Both isoforms are modified by phosphorylation, acetylation, S-nitrosylation, oxidation, methylation, sumoylation, and ubiquitination. Many of the modified residues are conserved between Hsp90α and Hsp90β. However, there are a few differences between the two, which allow for specific functions or regulation in an isoform-specific way. We refer the reader to a very recent review on this topic [65].

### 3.4. Evolutionary Divergence in Gene Expression

*HSP90AB1* evolved as the constitutively expressed isoform, while *HSP90AA1* evolved to be inducible in response to different types of stresses. The differential gene expression patterns of mammalian *HSP90AA1* and *HSP90AB1* were first characterized with transformed mouse cells [66]. Ullrich and colleagues showed that *HSP90AB1* is constitutively expressed under normal conditions and has a 2.5-fold higher expression level than *HSP90AA1*. However, upon heat shock, *HSP90AA1* expression increased 7-fold, while *HSP90AB1* increased only 4.5-fold [66]. This suggested that Hsp90β can also be induced when cells are under stress. In a recent study, we observed that Hsp90β expression can be induced by genetic stress as well as by long-term moderate heat stress at the translational level through an IRES in the Hsp90β mRNA [49]. Other studies also showed that Hsp90β can be induced under heat and nutrient stress [67,68]. Heat shock factor 1 (HSF1), which is recruited to DNA through heat shock elements (HSEs) and is the transcriptional master regulator of the response to heat shock and several other stresses, regulates the expression of Hsp90α [69,70]. Upstream of *HSP90AA1*, there are several heat shock elements (HSEs) which enable the stress-mediated expression of Hsp90α [71], and in particular, there is a distal HSE located at −1031 bp from the transcription start site (TSS), which is required for heat-shock induction [26]. Immediately upstream of the TATA box, the proximal HSE located at −96/−60 bp functions as a permissive enhancer. Another HSE is present within the first intron region at +228 bp from the TSS. In contrast, an upstream HSE of located at −648 bp of the TSS of *HSP90AB1* appears not to respond to heat shock [67]. However, HSEs located at +688/733 bp within the first intron are tightly bound by HSF1 and are important for maintaining the high constitutive and heat-shock induced expression levels [67]. In an apparent contradiction, it was observed for mouse oocytes that the basal level of *HSP90AB1* transcripts does not depend on HSF1 since *hsf1* knockout oocytes do not show any reduction in Hsp90β mRNA [72]. The constitutively active core promoter (−36 to +37 bp) of *HSP90AB1* has a CAAT box, a specificity protein 1 (SP1) binding site, and a TATA box (−27 bp) [71]. The promoter of *HSP90AA1* does not contain a CAAT box, but one has been identified far upstream at −1144 bp. The binding of Krüppel-Like-Factor 4 (KLF4) to the promoters of *HSP90AB1* and *HSP90AA1* leads to higher expression of both isoforms [73]. In addition to the activation by HSF1, Hsp90β is upregulated by the signal transducer and activator of transcription (STAT) family transcription factors [74]. Interferon-γ (IFN-γ) activation of STAT1 also induces Hsp90β expression [74]. For the induction of *HSP90AB1* by heat shock, STAT1 phosphorylation by Jak2 and PKC is necessary [75]. However, the activation of these kinases in turn requires the association with Hsp90, establishing a positive auto-regulatory loop. *HSP90AB1* expression is also regulated at the translational level by mTORC1 [76]. This seems to be dependent on a 5′-terminal oligopyrimidine motif in the 5′UTR of the Hsp90β mRNA, but the mechanism is not known. As a general negative feedback mechanism, the Hsp90 complex represses HSF1 activation, thereby inhibiting an over-activation and timely attenuation of the HSF1 response [77,78]. Clearly, there is still much to learn about the tissue- and cell type-specific regulation of the two Hsp90 isoforms, both at the transcriptional and translational levels.

### 3.5. Functional Specificities of the Two Isoforms

In humans, Hsp90α and Hsp90β together are predicted to interact with more than 2000 proteins [79]. Not quite that many have been experimentally validated, and in the vast majority, isoform specificity has not been thoroughly investigated. Regarding the Hsp90 interactome, we refer the reader to continuously updated resources, which we have been making available as a searchable online database at https://www.picard.ch/Hsp90Int, accessed on 5 August 2022 [79] and a downloadable file at https://www.picard.ch/downloads/Hsp90interactors.pdf, accessed on 5 August 2022. It is natural to assume that highly homologous versions of Hsp90s found in the same cellular compartment would have identical functions. However, these cytosolic Hsp90 isoforms have evolved to have some overlapping, synergistic, and distinct isoform-specific functions. In some contexts, they even have antagonistic functions (see below). Taipale and colleagues systematically characterized the chaperone/co-chaperone/client interaction network in human cells [80]. They provided evidence for both overlapping and distinct client specificities. When they analyzed isoform-specific interactors by gene ontology terms, this again translated to both common and isoform-specific terms. This generally supports the conclusion that isoform-specific interactomes impart isoform-specific functions. It is worth mentioning here that a characterization of the interactomes of the two isoforms of *S. cerevisiae* by immunoprecipitation/mass spectrometry led to globally similar conclusions, except that it was noted that the vast majority of clients are shared by both isoforms [29].

In the next section of the discussion of isoform-specific functions and clinical relevance (Figure 2), we will consider those findings where the involvement of one specific isoform was experimentally validated by using isoform-specific antibodies and genetic or pharmacological inhibition. With many studies, it must be kept in mind, though, that it is not easily possible to exclude the possibility that the observed phenotype is due to reduced total Hsp90 levels or activity. Indeed, we know from our own studies that phenotypes can be due to the latter rather than the loss or inhibition of a specific isoform [49].

### 3.6. Hsp90β-Specific Functions

During the course of evolution, the *HSP90AB1* gene has evolved to be expressed more or less constitutively, presumably to support essential cellular housekeeping activities. The embryonic lethality of the mouse knockout may be a reflection of that [81]. Hsp90β was shown to play a role in trophoblast differentiation and that Hsp90β-deficient homozygous mouse embryos with normal expression of Hsp90α failed to differentiate to form placental labyrinths. This resulted in lethality beyond day 9 of embryonic development. While it could be speculated that this indicated a housekeeping role for Hsp90β, it was also suggested that the developmental arrest could be due to a defective critical and potentially Hsp90β-dependent client such as the bone morphogenetic protein receptor. Interestingly, Hsp90β was demonstrated to regulate the pluripotency of embryonic stem cells via regulating the transcription of Nanog through an interaction with STAT3 [82]. Thus, whether mammalian development truly depends specifically on Hsp90β for the aforementioned reasons or whether it depends on a threshold level of total Hsp90 remains to be determined.

Hsp90β appears to have an exclusive role in muscle cell differentiation and regeneration in the mouse [50,83]. We showed that during skeletal muscle differentiation in the mouse, there is a unique Hsp90 isoform switch [50]. When mouse myoblasts differentiate into myotubes, Hsp90α disappears, and only Hsp90β remains. Hsp90β interacts with the muscle-specific Hsp90 co-chaperone Aarsd1L to support the differentiation of myotubes. As these differentiate, Aarsd1L replaces the ubiquitous cochaperone p23. Later, He and colleagues discovered the importance of Hsp90β in muscle regeneration after tissue injury [83]. They found that in a mouse muscle injury model, the Hsp90β isoform, but not Hsp90α, was strongly elevated during the first few days post injury. Hsp90β expression levels normalized when active myogenesis eventually ceased. Following muscle injury, p53-dependent persistent senescence impairs muscle repair. During regeneration, Hsp90β interacts with the p53-inhibitory protein MDM2 to suppress p53-dependent senescence of the injured muscle. Degeneration of skeletal muscle is one of the features of aging in humans [84]. The reduction of quiescent muscle stem cells through senescence leads to the decline in muscle regeneration in aged mice [85]. Hence, enhancing Hsp90β activity might be protective for muscle fibers during aging.

Hsp90β is involved in controlling the formation of endodermal progenitor cells and development of the liver [86]. For hepatocyte formation, the transcription factor hepatocyte nuclear factor 4 α (HNF4α) is essential [87]. Hsp90β interacts with HNF4α to regulate its half-life and is thus directly linked with the formation of hepatocytes from progenitor cells. The liver is the primary organ involved in the metabolism of nutrients. Not surprisingly, the specific function of Hsp90β in liver formation further connects it to different metabolic disorders. Hsp90β is involved in glucose and cholesterol metabolism [68,88]. In human skeletal muscle myoblasts and in a mouse model of diet-induced obesity, Hsp90β was found to regulate glucose metabolism and insulin signaling. Studies showed that the knockdown of Hsp90β improves glucose tolerance, alters the expression of key metabolic genes, and enhances the activity of the pyruvate dehydrogenase complex. Furthermore, Hsp90β is essential for lipid homeostasis by regulating fatty acid and cholesterol metabolism [88]. Depleting Hsp90β promotes the degradation of mature sterol regulatory element-binding proteins through the Akt-GSK3β-FBW7 pathway, and hence decreases the content of neutral lipids and cholesterol in the body [88,89].

Another important isoform-specific function of Hsp90β is regulating the responsiveness to vitamin D [90]. In the intestine, enterocytes require Hsp90β for optimal vitamin D responsiveness by regulating vitamin D receptor (VDR) signaling. It was observed that knocking down Hsp90β led to reduced vitamin D-mediated transcriptional activity. It is noteworthy that VDR is a member of the nuclear receptor family of transcription factors, which comprises some of the most prototypical Hsp90 clients, such as the steroid receptors.

Hsp90β is necessary for maintaining the neuromuscular junction (NMJ) [91]. Rapsyn, an acetylcholine receptor-interacting protein, is essential for synapse formation [92]. Hsp90β is necessary for rapsyn stabilization and regulating its proteasome-dependent degradation. Luo and colleagues showed that inhibition of Hsp90β activity or expression or disruption of its interaction with rapsyn impairs the development and maintenance of the NMJ.

### 3.7. Targeting Hsp90β in Different Diseases

As alluded to above, Hsp90β influences pathways regulating insulin resistance [68]. It was observed that when Hsp90β was inhibited, blood glucose levels were reduced. Thus, targeting Hsp90β might help to regulate the blood sugar of patients with type 2 diabetes. In patients with nonalcoholic fatty liver disease (NAFLD), the Hsp90β levels in serum was found to be very high [93]. Balanescu and colleagues conducted a study on overweight and obese children and found serum Hsp90β, but not Hsp90α, to be significantly higher. This suggests that the ratio of Hsp90α and Hsp90β in blood serum could be a prognostic biomarker for NAFLD. Jing and colleagues found that the novel Hsp90β-selective inhibitor corylin significantly reduced lipid content in both liver cell lines and human primary hepatocytes [88]. In animal models, they observed that corylin ameliorated NAFLD, type 2 diabetes, and atherosclerosis.

Reduced levels of both Hsp90α and Hsp90β are associated with neuronal cell death in patients suffering from Alzheimer’s disease (AD) [94]. However, Zhang and colleagues showed that Aβ-induced stress decreased the levels of Hsp90β, but not Hsp90α. Reduced levels of Hsp90β were strongly correlated with reduced abundance of its client and nuclear receptor PPARγ, and down-regulated Aβ clearance-related genes in primary microglia [95]. This exciting observation led them to think about increasing Hsp90β levels in the AD mouse model. Using the natural compound jujuboside A (JuA), they observed that it significantly restored the content and function of PPARγ by enhancing the expression of Hsp90β. JuA-treated AD mice displayed ameliorated cognitive deficiency. In a recent study, Wan and colleagues showed reduced levels of both Hsp90α and Hsp90β levels in the hippocampal CA3 region of the APP/PS1 mouse model of Alzheimer’s disease [96]. However, the overexpression of Hsp90β, but not Hsp90α, ameliorated neuronal and synaptic loss, suggesting Hsp90β has a specific neuroprotective role. High-dose preventive treatment with erythropoietin (EPO) attenuated Aβ-induced astrocytosis and increased neovascularization in the hippocampus of the mouse AD model. It reversed dendritic spine loss via upregulation of Hsp90β. Therefore, inducing Hsp90β expression might be explored for the treatment of AD patients.

Hsp90β enhances the innate immune response [97]. Hsp90β interacts with the protein stimulator of interferon genes (STING) and stabilizes STING protein levels in response to microbial infections, allowing the activation of the downstream target TBK1, which is itself an Hsp90 interactor, for inducing IFN responses. This suggests inducing Hsp90β could also be efficient against DNA viruses and microbial infections. Sato and colleagues showed that reduced Hsp90β levels are associated with infections with Herpes simplex virus-1 (HSV-1) and *Listeria monocytogenes*, suggesting that boosting Hsp90β levels and/or activity may protect against pathogenic infections. If so, EPO might also be beneficial against bacterial and viral infections.

Although it is predominantly Hsp90α that is overexpressed in different types of cancer, in some cancers it is Hsp90β, which appears to be responsible for cancer cell survival [98,99,100,101,102]. As discussed above, there are tissues that rely primarily on Hsp90β. It appears that cancers of those tissues often maintain this dependence. For example, hepatocytes primarily require Hsp90β, as do hepatocellular carcinoma cells, notably for vascular endothelial growth factor receptor (VEGFR)-mediated angiogenesis [99,100]. In other cancers, Hsp90α regulates VEGFR-mediated angiogenesis. Meng and colleagues evaluated angiogenesis in hepatocellular carcinoma upon knockdown of Hsp90α or Hsp90β, inhibiting the remaining isoform with an Hsp90 inhibitor. They observed that VEGFR-mediated angiogenesis was inhibited by an Hsp90β inhibitor. Heck and colleagues showed that selective Hsp90β inhibition in human myeloid leukemia cells results in apoptosis [103]. Hsp90β-apoptosome interactions also contribute to chemoresistance in leukemias [104]. Hsp90β inhibition could kill leukemia cells by promoting the degradation of the Hsp90 client HIF1α [103]. Heck and colleagues treated cells with the Hsp90α-selective inhibitor KUNA110, the Hsp90β-selective inhibitor KUNB105, or the pan-Hsp90 inhibitor 17AAG. Inhibition of Hsp90α did not trigger cell death. However, Hsp90β inhibition led to cell death by TNFα- and TRAIL-induced HIF1α degradation. HIF1α is an interactor of both Hsp90 isoforms (see https://www.picard.ch/Hsp90Int, accessed on 5 August 2022), and yet the inhibition of Hsp90β, but not Hsp90α, led to the degradation of HIF1α in leukemia cells. This surprising result illustrated the potential of Hsp90 isoform-specific inhibition for the treatment of certain types of cancer.

In Ewing’s sarcoma, it was found that Hsp90β inhibition leads to decreased expression of the multidrug resistance-associated protein 1 associated with mitochondria [105]. In laryngeal carcinoma (LC), Hsp90β directly interacts with Bcl-2 and is involved in the anti-apoptotic progression of LC [101]. In osteosarcoma, extracellular Hsp90β secreted by MG63 cells was found to be associated with cancer cell survival [106]. This could be connected with the observation that muscle cells in the mouse are solely dependent on Hsp90β for differentiation and regeneration. This evidence collectively suggests that sarcoma is mainly dependent on Hsp90β [107]. Not only that, even in the skeletal muscle disorder myotonia congenita, Hsp90β plays an essential role in the quality control of the chloride channel CLC-1 by dynamically coordinating protein folding and degradation. Peng and colleagues showed that by using Hsp90β inhibitors, CLC-1 degradation in myotonia patients can be prevented [108].

Hsp90β plays a role in drug resistance in lung cancer. The P-glycoprotein (P-gp) encoded by the *MDR1* gene is responsible for exporting drugs from cells. Kim and colleagues showed that casein kinase 2 (CK2)-mediated phosphorylation of Hsp90β and the subsequent stabilization of its client PXR, a nuclear receptor, is a key mechanism in the regulation of MDR1 expression [109]. Inhibition of both CK2 and Hsp90β enhances the down-regulation of PXR and P-gp expression. High level expression of Hsp90β is also associated with poor survival in resectable non-small-cell lung cancer patients [98].

Possibly related to the dependence of liver cells on Hsp90β discussed above, Hepatitis B virus (HBV), which causes chronic infection in the liver, evades the immune defense by interaction with Hsp90β [110]. Hsp90β inhibition could also be a useful therapeutic approach in *Helicobacter pylori-*induced gastric injury [111]. Cha and colleagues showed that Hsp90β physically interacts with Rac1, which resulted in the activation of NADPH oxidase. NADPH oxidase activation leads to the production of ROS and increased inflammation in infected cells. Suppression of *H. pylori*-induced translocation of Hsp90β to the membrane may ameliorate gastric injury. Nickel ions-mediated inflammation also occurs through Hsp90β in human B-cells. Nickel ions bind to the linker domain of Hsp90β and reduce its interaction with HIF1α. The released HIF1α then becomes more localized in the nucleus and enhances IL-8 expression [112].

### 3.8. Hsp90α-Specific Functions

The stress-inducible isoform Hsp90α helps cells adapt to stress [113]. Most of the functions of Hsp90α are thus connected to stress response pathways and proteins. Unlike Hsp90β, Hsp90α is not necessary for viability in the mouse [49,59]. In normal conditions, as discussed above, some organs have a high abundance of Hsp90α, whereas others have negligible expression levels (https://www.proteinatlas.org/search/HSP90AA1; accessed on 5 August 2022). For example, the brain has the highest levels of Hsp90α mRNA expression, which is several folds higher than any other organs. However, the highest levels of mRNA are not translated into proteins as brain expresses only moderate amounts of Hsp90α protein. A recent study says the human brain has a higher temperature than the usual body temperature ranging from 36.1 to 40.9 °C in a circadian way [114]. In addition, the brain temperature varies by age, sex, menstrual cycle, and brain region. It is conceivable that such temperature increases in the brain might trigger a heat-shock response, temporarily generating more Hsp90α protein from the already elevated levels of Hsp90α mRNA. Since brain cells face a substantial temperature fluctuation, proteins may be more prone to misfolding, and brain cells may therefore need higher amounts of Hsp90α under certain circumstances. If we consider reproductive organs, testis and fallopian tube have high levels of Hsp90α expression. At the cellular level, basal prostate cells have the highest amount of Hsp90α mRNA among all cell types. Spermatocytes also have high Hsp90α. This suggests reproductive organs require Hsp90α. Earlier, our lab established that Hsp90α is required for male fertility in mice [59]. Mice without Hsp90α can survive normally but are sterile because of a complete failure to produce sperm. Interestingly, Hsp90α knockout mice develop normal reproductive organs, but spermatogenesis specifically arrests at the pachytene stage of meiosis I. Supporting these findings, Kajiwara and collogues later complemented these findings by demonstrating that spermatogenesis also arrests when the Hsp90α gene is conditionally deleted at the adult stage [115]. Intriguingly, Hsp90α controls the biogenesis of fetal PIWI-interacting RNAs, which act against endogenous transposons during the development of male germ cells in mammals [116]. The Hsp90α knockout causes a reduction of HIF1α levels in the testis, which may also contribute to blocking sperm production and causing infertility [117]. The downregulation of Hsp90β had little effect on the hypoxia-induced accumulation of HIF1α. Thus, HIF1α is required for proper spermiogenesis, and it is the Hsp90α isoform that is needed to keep HIF1α functional, even though both Hsp90α and Hsp90β can interact with HIF1α [118,119]. It remains to be seen whether this unique Hsp90α role has anything to do with the specific physiology and temperature-sensitivity of the testis.

Oogenesis may also be largely dependent on the Hsp90α isoform. Metchat and colleagues showed that extremely low levels of Hsp90α correlate with the developmental defects of *hsf1*-/- oocytes [72]. While *hsf1*-/- females produce oocytes, they do not carry viable embryos. However, later we showed that no difference in embryo production was observed in female mice lacking Hsp90α compared to the wild-type [49,59]. Hence, it is possible that *hsf1*-/- oocytes failed to develop because of some Hsp90α-independent issue.

In the human retina, rod cells have high levels of Hsp90α mRNA expression (https://www.proteinatlas.org; accessed on 5 August 2022), which may be related to elevated local temperatures upon exposure to light. The local rise in temperature of human retina exposed to direct sunlight is about 2 °C [120]. It may even tolerate a local rise of at least 10 °C [121], and yet, intense light causes thermal damage [122]. Wu and colleagues found that Hsp90α deficiency in mice could lead to retinitis pigmentosa [48], a common inherited retinal disease involving progressive photoreceptor degeneration and eventually blindness. They observed that both Hsp90α and Hsp90β were expressed in the developing retina of neonatal mice. Once the retina was fully developed, Hsp90α became the major Hsp90 isoform. In retinal photoreceptors, Hsp90α deficiency caused Golgi apparatus disintegration and impaired intersegmental vesicle trafficking. A proteomic analysis identified the microtubule-associated protein 1B (MAP1B) as an Hsp90α-associated protein in photoreceptors. Hsp90α deficiency increased the degradation of MAP1B by inducing its ubiquitination, causing α-tubulin deacetylation and microtubule destabilization, all potentially contributing to photoreceptor degeneration.

Muscle usually does not have much Hsp90α expression. However, among different type of muscle cells, cardiomyocytes have comparatively high Hsp90α mRNA expression (https://www.proteinatlas.org; accessed on 5 August 2022). Peterson and colleagues showed that the potassium channel hERG, which is critical for cardiac repolarization, solely interacts with Hsp90α and not with Hsp90β [123]. They found a direct relationship between Hsp90α and trafficking of hERG. Hence, the negative impact on hERG and the resulting cardiotoxicity must be considered in the context of treatments with pan-Hsp90 or with Hsp90α-specific inhibitors.

The DNA damage response is assisted by Hsp90α. The DNA-dependent protein kinase (DNA-PK) is a component of the DNA repair machinery, and it is a client of both Hsp90α and Hsp90β [124,125]. However, it was shown that Hsp90α is involved in DNA-PK-mediated DNA repair and apoptosis, but not Hsp90β [126,127]. Hsp90α itself is phosphorylated by DNA-PK at threonines 5 and 7 within its unique N-terminal sequence. Quanz and colleagues found that DNA damage induces the phosphorylation of Hsp90α at the aforementioned sites and its accumulation at sites of DNA double-strand breaks (DSB), where it associates with repair foci and promotes DNA repair [126]. Solier and colleagues showed that phosphorylated Hsp90α is located in the “apoptotic ring” upon induction of apoptosis. Although both phenomena are mediated by DNA-PK, Hsp90α phosphorylation is markedly greater and faster in response to apoptosis than to DNA damage [127]. An additional connection to the DNA damage response comes from the identification of the DNA damage response proteins NBN, and the ataxia-telangiectasia mutated kinase as Hsp90α clients [128]. It is conceivable that Hsp90α-specific inhibition would lead to their destabilization, contributing to defective DNA damage signaling, impaired DNA DSB repair, and increased sensitivity to DNA damage.

Hsp90α controls addictive behavior through the μ opioid receptor (MOR) [129]. Previously, Hsp90 had been found to be required for opioid-induced anti-nociception in the brain by promoting MAPK activation [130]. 17-AAG, a non-selective Hsp90 inhibitor, reduced opioid anti-nociception. In an independent study by Zhang and colleagues, treatment with 17-AAG was observed to reduce morphine analgesia, tolerance, and dependence in mice [131]. Interestingly, Lei and colleagues later found that specific inhibition of Hsp90α with the Hsp90α-selective inhibitor KUNA115 strongly blocked morphine anti-nociception in mice. In contrast, specific inhibition of Hsp90β with the inhibitor KUNB106 did not have any effect on morphine anti-nociception. Their observation suggests that Hsp90β is not involved in regulating opioid anti-nociception in the mouse brain. Surprisingly, Zhang and colleagues demonstrated by co-immunoprecipitation that Hsp90β and not Hsp90α associated with MOR in HEK293T and SH-SY5Y cells [131]. 17-AAG blocked the Hsp90β-MOR interaction and compromised MOR signal transduction in mice. For now, these findings remain contradictory, but if the Hsp90α-specific character of this function could be confirmed, it would suggest the possibility of using Hsp90α inhibitors in psychiatric patients with substance addiction.

Besides functioning as an intracellular molecular chaperone, Hsp90α is also secreted from cells [132,133,134,135]. All cells appear to secrete Hsp90α (eHsp90α) in response to environmental stress signals, including heat, hypoxia, inflammatory cytokines, ROS, oxidation agents, and several other stresses [134]. However, normal keratinocytes secrete eHsp90α only in response to tissue injury [136]. When skin is injured, keratinocytes massively release eHsp90α into the wound bed to promote wound repair [135,137]. Cheng and colleagues proposed that eHsp90α drives inward migration of the dermal cells into the wound, which is essential for wound remodeling and formation of new blood vessels [138]. Interestingly, this wound healing activity of eHsp90α does not require dimerization [136] nor ATPase activity, which is, of course, essential for chaperoning [138]. Instead, in this case, only a relatively small portion of eHsp90α is sufficient to elicit the response, essentially as a mitogen, through the LDL-receptor-related protein 1 (LRP1) [139].

### 3.9. The Clinical Relevance of Targeting Hsp90α

Specifically targeting the Hsp90α isoform could be an attractive therapeutic strategy for treating certain cancers [132,140]. Cancer cells are continuously under replicative, hypoxic, nutrient, and several other stresses [141,142]. Cellular stress leads to the upregulation of the inducible isoform Hsp90α. Hence, in most cancers, Hsp90α is highly upregulated. The knockdown of Hsp90α results in the degradation of several oncogenic client proteins, which suggests that the administration of an Hsp90α-selective inhibitor against Hsp90α-dependent cancers could be beneficial [143]. During cancer progression, many transcription factors encoded by proto-oncogenes are either stabilized by Hsp90α or induce the expression of Hsp90α. For example, the proto-oncogene MYC induces *HSP90AA1* gene expression [144]. The growth hormone prolactin induces *HSP90AA1* expression in breast cancer cells through STAT5 [145]. The nuclear factor-κB (NF-κB) stimulates anti-apoptotic pathways in cancer [146,147]. Ammirante and colleagues showed that two NF-κB putative consensus sequences are present in the *HSP90AA1* 5′ flanking region, and not in that of *HSP90AB1* [148]. This may explain why NF-kB-driven tumorigenic transformation leads to induced *HSP90AA1* expression. In head and neck cancer cells, the transcription factor SOX11 binds to HSP90α [149]. In breast cancer stem cells, Hsp90α and GRP78 interact with PRDM14 [150,151]. As discussed above, *HSP90AB1* is also induced in certain cancers. However, the stress-inducible gene *HSP90AA1* can be expressed several-fold higher than *HSP90AB1*. Thus, the balance of Hsp90α to Hsp90β is specifically shifted towards Hsp90α in cancer cells. It had been found that Hsp90α accounts for 2–3% of total cellular proteins in normal cells, but up to 7% in certain tumor cell lines [152]. Cancer cells may constitutively secrete Hsp90 [153,154,155], which is essential for enhancing their invasiveness [156]. Although Hsp90β can also be secreted by certain cells, it is eHsp90α and not eHsp90β that is required for invasion in a panel of cancer cell lines. eHsp90α activates matrix metalloproteinase-2, which may be one of the underlying mechanisms explaining enhanced invasiveness and metastasis of cancer cells [132,139,151,157]. The translocation of Hsp90α to the plasma membrane is stimulated by PLCγ1-PKCγ signaling [158], and by mutant p53 via Rab coupling protein-mediated Hsp90α secretion [159]. When Hsp90α is inhibited, the invasiveness of cancer decreases [160,161,162,163]. The plasma eHsp90α levels in patients with various cancers correlate with cancer stage [164,165,166]. For example, plasma Hsp90α levels were increased in patients with thymic epithelial tumor, hepatocellular carcinoma, and colorectal cancer [164,165,166,167]. This suggests that serum Hsp90α levels can be a prognostic marker in patients before and during treatment.

Hsp90α plays a significant role in idiopathic pulmonary fibrosis (IPF) [168]. Bellaye and colleagues showed how Hsp90α and Hsp90β synergistically promote myofibroblast persistence in lung fibrosis [169]. Hsp90α, but not Hsp90β, is secreted from IPF lung fibroblasts driven by tissue stiffness and mechanical stretch. Surprisingly, although Hsp90β is not secreted, it binds to LRP1 intracellularly, thus stabilizing the eHsp90 receptor and promoting LRP1 signaling, which feeds forward by inducing the secretion of Hsp90α. Inhibition of eHsp90α, which is increased in serum of patients with IPF, could be beneficial in treating IPF. The non-cell-permeable HSP90 inhibitor HS30 significantly inhibited eHSP90α and LRP1 colocalization, which was significantly increased in patients with moderate and severe IPF. In patients suffering from chronic obstructive pulmonary disease, Hsp90α levels were also found to be elevated in the serum [170], again suggesting that eHsp90α could be used as a biomarker of disease progression. The dysfunction of the airway epithelial barrier is closely related to the pathogenesis of asthma, and eHsp90α participates in the inflammation in asthma [171]. House dust mites (HDM) induce a dysfunction of the airway epithelial barrier. Mice with HDM-induced asthma have high levels of eHsp90α in bronchoalveolar lavage fluid and serum, and eHsp90α can cause the broncial epithelial hyperpermeability. 1G6-D7, a highly selective and inhibitory antibody against Hsp90α, was found to protect against HDM-induced airway epithelial barrier dysfunction. This suggests that eHsp90α-targeted therapy might be a potential asthma treatment.

Wang and colleagues found that HSV-1 survive inside cells using Hsp90α of the host. Hsp90α stabilizes the virion protein 16 (VP16) and promotes VP16-mediated transactivation of HSV-α genes [172]. When Hsp90α was knocked down or inhibited pharmacologically, it resulted in reduced levels of VP16 and of proteins encoded by the HSV-α genes. Considering that Hsp90β may have opposite effects on HSV-1 infections, since they are associated with a drop in Hsp90β levels (see above), careful investigations with highly Hsp90 isoform-selective inhibitors are clearly warranted in order to develop Hsp90-based therapies.

Loss-of-function mutations in the gene encoding the voltage-gated potassium channel KCNQ4 cause DFNA2, a subtype of autosomal dominant non-syndromic deafness characterized by progressive sensorineural hearing loss. The knockdowns of the two Hsp90 isoforms had opposite effects on the total KCNQ4 levels [173,174]. Specifically, the knockdown of Hsp90β led to a dramatic decrease, while the knockdown of Hsp90α resulted in a marked increase. Consistent with these results, overexpression of Hsp90β increased the KCNQ4 levels, whereas up-regulation of Hsp90α expression decreased the total KCNQ4 levels. This suggests that a combination of Hsp90α inhibitor and Hsp90β activator could potentially treat DFNA2.

Hsp90α also has a connection with diabetes. High glucose was shown to induce the translocation of Hsp90α to the outside of aortic endothelial cells [175]. In high glucose conditions, phosphorylation of Hsp90α was increased in a manner dependent on cAMP/protein kinase A, which was responsible for the membrane translocation of Hsp90α and reduced endothelial nitric oxide synthase (eNOS) activity [176]. eNOS is responsible for the production of most of the vascular NO, deficiency in which can promote atherogenesis [177]. Further support for a role of Hsp90α in atherosclerosis and diabetes came from the finding that the levels of eHsp90α were upregulated in patients with aggravated diabetic vascular disease [178]. eHsp90 recruits monocytes through LRP1 activation, which indicates a connection between Hsp90α and inflammatory damage in diabetic vascular complications. These observations suggest that Hsp90α inhibition may be useful in treating patients with type 2 diabetes.

In contrast, the connection of Hsp90 with eNOS was found to be protective against the damages caused by ischemia-reperfusion by reducing the blood flow and glomerular filtration rate [179]. When there is renal ischemia, more Hsp90 is beneficial [179]. Intra-renal transfection of expression plasmids for either Hsp90 isoform was shown to be protective. The protective effect was associated with restoring eNOS–Hsp90 coupling, reestablishing normal PKCα levels, and reducing Rho kinase expression. The transfection events were able to return eNOS phosphorylation to its basal state, restoring NO production and preventing reduced renal blood flow. Hsp90α is also a potential serological biomarker of acute rejection after renal transplantation [180]. Serum Hsp90α levels were significantly higher in kidney recipients upon rejection. In mice receiving a skin transplantation, serum Hsp90α was also found to be elevated when the first graft was rejected, and the levels further increased during more severe rejection of the second graft.

Elevated serum Hsp90α had been found in nonalcoholic steatohepatitis. Serum Hsp90α was increased in patients with metabolic-associated fatty liver disease (MAFLD). A positive correlation was found between age, glycosylated hemoglobin, serum Hsp90α, and grade of steatohepatitis [181]. Xie and colleagues showed that in a MAFLD mouse model, treatment with geranylgeranylacetone leads to decreased Hsp90α levels followed by improvement of steatohepatitis. To the extent that MAFLD may be the same as NAFLD, for which Hsp90β had been pinpointed (see above), here too, a careful classification of MAFLD/NAFLD patients with respect to clinical parameters and Hsp90 levels will be necessary before any Hsp90-based therapy can be considered.

## 4. Future Perspectives and Conclusions

Overall, Hsp90α-dependent processes contribute to stress adaptation or other specialized functions, while Hsp90β is essential for maintaining standard cellular functions such as cell viability. We recently demonstrated that at cellular and tissue levels, albeit with some exceptions, it is the total Hsp90 levels that matter to sustain essential basic functions, without overt isoform-specific requirements [49]. Pan-Hsp90 inhibitors affect a broad range of key cellular processes, which may have contributed to the failure of several Hsp90 inhibitors in clinical trials [182,183]. However, from the evidence presented in this review, it appears that there are indeed physiological and pathological conditions where one particular isoform is more involved than the other. Hence, targeting only the one critical isoform with isoform-specific inhibitors is the way to go for safer and more efficient treatments (Table 2, Table 3 and Table 4). Towards reaching that ultimate goal, several major challenges remain. More insights into organ- and cell type-specific functions of the Hsp90 isoforms are needed to stratify patients appropriately for isoform-targeted treatments. Although several groups have begun to report the discovery of isoform-selective inhibitors (Table 5 and Table 6), there is still a lot of room for improvement. The ideal Hsp90 isoform-specific inhibitor would have the following features: (i) High isoform-selectivity or even -specificity; (ii) high Hsp90 specificity with limited effects on other biomolecules; (iii) drug-like characteristics, i.e., have favorable pharmacokinetics and pharmacodynamics; (iv) oral availability; (v) for some applications, the ability to cross the blood–brain and blood–testis barriers.

## Figures and Tables

**Figure 1 biomolecules-12-01166-f001:**
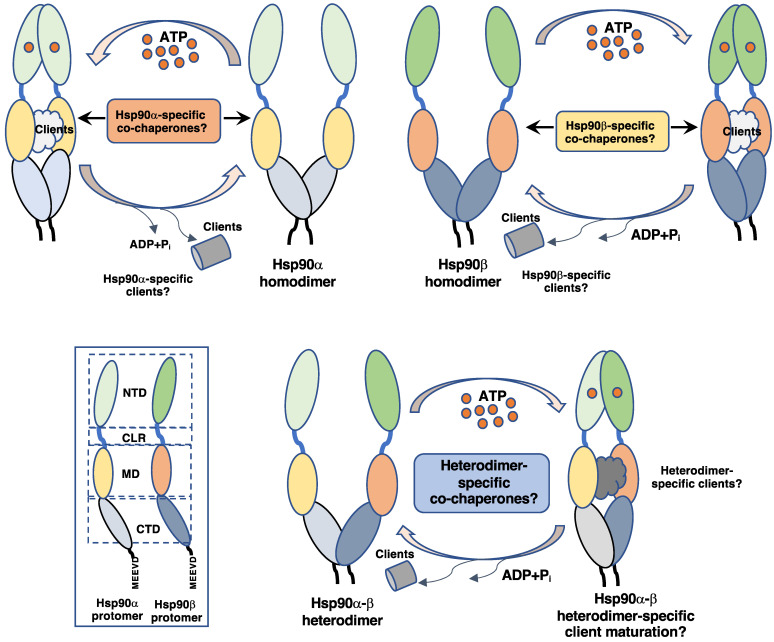
Schematic representation of the molecular chaperone cycle of Hsp90α and Hsp90β, either as isoform homodimers or hypothetically as isoform heterodimers. NTD, MD, and CTD, N-terminal, middle, and C-terminal domains, respectively; CLR, charged linker region.

**Figure 2 biomolecules-12-01166-f002:**
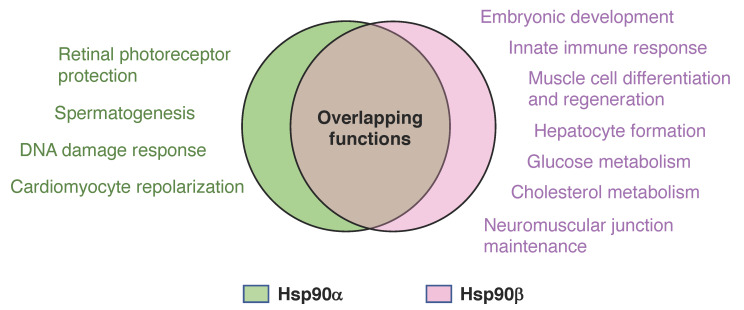
Venn diagram of common and isoform-specific functions of Hsp90. Isoform-specific functions, as discussed in the text, are highlighted in the corresponding colors.

**Table 1 biomolecules-12-01166-t001:** Some co-chaperones of Hsp90α and Hsp90β ^1^.

Co-Chaperones	Function/Comments
Aha1	Accelerator of Hsp90 ATPase
Hop	Adaptor between Hsp70 and Hsp90; inhibitor of Hsp90 ATPase
p23	Binds closed Hsp90 conformation, inhibits ATPase
Cdc37	Kinase-specific co-chaperone
FKBP51/52	Peptidylprolyl-cis/trans-isomerase; maturation and activation of steroid receptors
Cyp40	Peptidylprolyl-cis/trans-isomerase
PP5	Phosphatase interacting with Hsp90
CHIP	E3 ubiquitine ligase
Pih1	Component of the Rvb1-Rvb2-Tah1-Pih1 (R2TP) complex
Tah1	Component of the Rvb1-Rvb2-Tah1-Pih1 (R2TP) complex
TTC4	Genetic interaction with Cpr7; regulator of protein translation
FKBP8	Peptidylprolyl-cis/trans-isomerase; may preferentially bind Hsp90β
UNC45A	Preferentially binds Hsp90β
Aarsdl1	Competes with p23; only binds Hsp90β

^1^ Only some of the most frequently investigated co-chaperones are listed and some of those with reported Hsp90 isoform-selectivity. For full list of co-chaperones and references, see https://www.picard.ch/downloads/Hsp90facts.pdf.

**Table 2 biomolecules-12-01166-t002:** Role of specific Hsp90 isoforms in diseases ^1^.

Expression Levels	Disease
Higher levels of Hsp90α	Idiopathic pulmonary fibrosis Asthma Autosomal dominant non-syndromic deafness Diabetes type 2 Nonalcoholic steatohepatitis
Lower levels of Hsp90α	Male infertility
Higher levels of Hsp90β	Nonalcoholic fatty liver disease
Lower levels of Hsp90β	Aβ-induced Alzheimer’s disease DNA viruses and microbial infections

^1^ See text for details and references.

**Table 3 biomolecules-12-01166-t003:** Cancers with upregulation of specific Hsp90 isoforms ^1^.

Cancers with Higher Levels of Hsp90β	Cancers with Higher Levels of Hsp90α
Sarcoma	Breast cancer
Hepatocellular carcinoma	Head and neck cancers
Myeloid leukemia	Epithelial cancer
Lung cancer	Colorectal cancer

^1^ See text for details and references.

**Table 4 biomolecules-12-01166-t004:** Diseases and hypothetical isoform-specific treatments ^1^.

Diseases	Hypothetical Therapy
Nonalcoholic fatty liver	Hsp90β inhibition
Aβ-induced Alzheimer’s disease	Hsp90β induction ^2^
Hepatocellular carcinoma	Hsp90β inhibition
Myeloid leukemia cells	Hsp90β inhibition
Ewing’s sarcoma	Hsp90β inhibition
Lung cancer	Hsp90β inhibition
Myotonia	Hsp90β inhibition
Hepatitis B virus infection	Hsp90β inhibition
*Helicobacter pylori-* induced gastric injury	Hsp90β inhibition
Opioid addiction	Hsp90α inhibition
Different cancers	Hsp90α inhibition
Idiopathic pulmonary fibrosis	eHsp90α inhibition
Herpes simplex virus-1 infection	Hsp90α inhibition
Autosomal dominant non-syndromic deafness	Hsp90α inhibitionHsp90β induction
Renal ischemia	Hsp90β/Hsp90α induction

^1^ See text for details and references. ^2^ “Induction” is meant to indicate either increased expression or increased activity.

**Table 5 biomolecules-12-01166-t005:** Isoform-specific inhibitors of Hsp90.

Compound	Hsp90 Isoform	Binding Site	References
KUNB31	Hsp90β	N-terminal domain	[184]
Vibsanin B and its derivatives	Hsp90β > Hsp90α	C-terminal domain	[185]
Corylin	Hsp90β	Amino acids 276–602 crucial for corylin binding	[88]
1G6-D7 (antibody)	eHsp90α	Fragment of 115 amino acids encompassing parts of charged and middle domains	[186]
HS30	eHsp90α	N-terminal	[187,188]
KU675	Hsp90α	C-terminal	[189]
NVP-BEP800	Hsp90β > Hsp90α	N-terminal	[190]

**Table 6 biomolecules-12-01166-t006:** Inducers of Hsp90β expression.

Compound	References
Jujuboside A	[95]
Erythropoetin	[96]

## Data Availability

Not applicable.

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
