# Peer review of "Cytosolic Hsp90 Isoform-Specific Functions and Clinical Significance"

_biomolecules, 2022, doi:10.3390/biom12091166_

Round 1
Reviewer 1 Report
This is a thorough and wide-ranging review of an emerging and important topic. The authors provide an excellent background and clearly discuss what needs to be done moving forward. I have no suggestions for further changes/modifications.
Author Response
We thank the reviewer for his/her positive response.
Reviewer 2 Report
Cytosolic Hsp90 isoform-specific functions and clinical significance
By Maiti, S., and Picard, D.
This manuscript is a comprehensive, yet concise review on the common and specific aspects of functions of two isoforms, alpha and beta, of the heat shock/stress protein Hsp90. The authors focus mostly on mammalian Hsp90. Isoform-specific functions and clinical significance of Hsp90 in many human diseases are precisely described. The section on the isoform-specific post-translational modification of Hsp90 is rather too short, however, this is justified by referring a very recent review on this topic. Sufficient amounts of proper references are included. The text is crisp and well-written in straight English. Overall, I believe this manuscript is quite appropriate for publication in the special review issue “Hsp90 Structure, Mechanism and Disease”. Minor issues and typo errors listed below should be corrected.
Title: This may be a matter of taste, but there are a mitochondrial and an ER-type forms of Hsp90s in mammals, therefore, the title might be better to be “Isoform-specific functions of cytosolic Hsp90 and clinical significance”.
Page 1, line 34: “... nomenclature committee)” -> nomenclature committee
Page 2, line 63: “... in only 27 amino acids” -> It would be better to show the total amino acid numbers of alpha and beta Hsp90 here.
Figure 1: The illustration may cause a misunderstanding that ATP binds to the very N-terminal end of Hsp90, therefore, it would be better to put the ATP circles in the middle of the NTD.
Page 3, line 90: “... and CTDN” -> and CTD
Page 5, line 155: “nutrient stress [67, 68]..” -> nutrient stress [67, 68].
Page 7, line 222: “HDuring regeneration” -> During regeneration
Page 8, line 276: “an Hsp90 interactors” -> an Hsp90 interactor
Page 10, line 357: “... some Hsp90alpha-independent issue” -> incomplete sentence
Page 12, line 432: “it iseHsp90alpha” -> it is eHsp90
Page 12, line 451: “IMice with ...” -> Mice?
Figure 2: The letter spacing of this figure in the PDF file looks inappropriate. Adjust the letter spacing or choose another font for better looking.
//End.
Author Response
Title: This may be a matter of taste, but there are a mitochondrial and an ER-type forms of Hsp90s in mammals, therefore, the title might be better to be “Isoform-specific functions of cytosolic Hsp90 and clinical significance”.
We appreciate this suggestion and have carefully considered this version and other variations. In the end, we decided to keep the original one, which very subtly "smells" more like what we wanted to emphasize in our review article.
Page 1, line 34: “... nomenclature committee)” -> nomenclature committee
corrected
Page 2, line 63: “... in only 27 amino acids” -> It would be better to show the total amino acid numbers of alpha and beta Hsp90 here.
Good point. We have added the size as well.
Figure 1: The illustration may cause a misunderstanding that ATP binds to the very N-terminal end of Hsp90, therefore, it would be better to put the ATP circles in the middle of the NTD.
Excellent point. We have revised the figure accordingly.
Page 3, line 90: “... and CTDN” -> and CTD
Page 5, line 155: “nutrient stress [67, 68]..” -> nutrient stress [67, 68].
Page 7, line 222: “HDuring regeneration” -> During regeneration
Page 8, line 276: “an Hsp90 interactors” -> an Hsp90 interactor
Page 10, line 357: “... some Hsp90alpha-independent issue” -> incomplete sentence
Page 12, line 432: “it iseHsp90alpha” -> it is eHsp90
Page 12, line 451: “IMice with ...” -> Mice?
We thank the reviewer for his/her very careful reading. We have corrected all afore-mentioned errors.
Figure 2: The letter spacing of this figure in the PDF file looks inappropriate. Adjust the letter spacing or choose another font for better looking.
Unfortunately, we are unable to reproduce the issue that the reviewer seems to have noticed in his/her copy. We have double checked it with both Preview and Acrobat Reader (and of course Word).
Reviewer 3 Report
The review article summarizes current knowledge on the function of Hsp90 isoform-specific functions and clinical significance. The topic selection is close to the forefront and attractive. The content is substantial, detailed, and in-depth for readers. While I find the review convincing and interesting, I have a few comments and questions.
1. The first suggestion is the authors mainly focus on mammalian Hsp90 and its cytosolic isoforms, I suggest that other organisms, such as HSP90 from bacteria, viruses, parasites, etc., can also be briefly introduced for their structure and function. For example, Schistosoma japonicum derived HSP90α (Sjp90α) stimulated the expression of dendritic cells and Th 17 cell response. I think this
2. Also, the authors have focus on the role of Hsp90 in the innate response, I suggested it should add some research about the Hsp90 in the adaptive immune response, such as Hsp90 inhibition ameliorates CD4+ T cell-mediated acute Graft versus Host disease in mice (PMID: 27980780). In addition, other study had found HSP90α on the surface of TRAPs programs the immunosuppressive functions of CD4+ T cells to promote tumor growth and metastasis (PMID: 31300052).
3. In recent years, some research suggested HSP90 as one target for COVID-19 therapy (PMID: 33585804), as inhibition of the HSP90 activity resulted in a reduction of viral replication and pro-inflammatory cytokine expression in primary human airway epithelial cells. I think this is very interesting. Suggest add it.
4. If possible, the authors could add the graph about the role of Hsp90 in different disease, making it easier for readers to grasp the disease function of HSP90.
Author Response
- The first suggestion is the authors mainly focus on mammalian Hsp90 and its cytosolic isoforms, I suggest that other organisms, such as HSP90 from bacteria, viruses, parasites, etc., can also be briefly introduced for their structure and function. For example, Schistosoma japonicum derived HSP90α (Sjp90α) stimulated the expression of dendritic cells and Th 17 cell response. I think this
This does not appear to have anything to do with Hsp90 isoforms (in particular of humans), which is the focus of our review. Schistosoma probably only has one cytosolic isoform and why the authors decided to refer to it as a is a mystery.
- Also, the authors have focus on the role of Hsp90 in the innate response, I suggested it should add some research about the Hsp90 in the adaptive immune response, such as Hsp90 inhibition ameliorates CD4+ T cell-mediated acute Graft versus Host disease in mice (PMID: 27980780). In addition, other study had found HSP90α on the surface of TRAPs programs the immunosuppressive functions of CD4+ T cells to promote tumor growth and metastasis (PMID: 31300052).
The fact that Hsp90 inhibition counteracts graft versus host disease is exciting, but unrelated to our discussion of cytosolic Hsp90 isoforms. Regarding Hsp90 on TRAPs, there is unfortunately no clear evidence that it is only Hsp90a since the authors used an antiserum we know cannot discriminate between the two isoforms.
- In recent years, some research suggested HSP90 as one target for COVID-19 therapy (PMID: 33585804), as inhibition of the HSP90 activity resulted in a reduction of viral replication and pro-inflammatory cytokine expression in primary human airway epithelial cells. I think this is very interesting. Suggest add it.
Obviously a great result, too, but again not related to our specific topic since the pan-Hsp90 inhibitor that these authors used does not allow conclusions about which isoform may be involved. In fact, it has been found by Kamel et al. (PMID: 34118193) that Hsp90b is also induced, suggesting that both isoforms may be involved, and of course, that Hsp90 inhibition in general may be worth looking at.
- If possible, the authors could add the graph about the role of Hsp90 in different disease, making it easier for readers to grasp the disease function of HSP90.
This is what Tables 3-4 are all about.